# Hospital at home: A systematic review of how medication management is conceptualised, described and implemented in practice—A study protocol

**Sophie McGlen[1], Clare Crowley[2], Daniel Lasserson[1,3], Zahra A. L. Qamariat[4], Rosemary H. M. Lim[2]***

**1** Oxford University Hospitals NHS Foundation Trusts, Oxford, United Kingdom, **2** Reading School of Pharmacy, University of Reading, Reading, Berkshire, United Kingdom, **3** Warwick Medical School, University of Warwick, Warwick, United Kingdom, **4** Pharmaceutical Affairs, Critical Care, Dammam Medical Complex, Eastern Health Cluster, Dammam, Saudi Arabia

* r.h.m.lim@reading.ac.uk

**Data Availability Statement:** No datasets were generated or analysed during the current study. All

## Abstract

### Introduction

Hospital at Home (H@H) is a method of healthcare delivery, where hospital level interventions are conducted in the patient's usual place of residence, offering an alternative to hospital admission. This often includes the ability to perform point of care diagnostics and treat conditions using a range of treatments traditionally associated with hospital admission, including intravenous medicines and oxygen. H@H services have been established worldwide but there is a wide variation in definition and delivery models and currently no documented evidence supporting the delivery of medicines and medicines management within the H@H model. Therefore, this study aims to 1) describe how medication management in H@H is conceptualised, 2) describe and identify key components of medication management in H@H and 3) describe and identify variability in the implementation of medication management services within H@H models.

### Methods and analysis

We will search a range of databases (PubMed, Medline, Embase, CINAHL), publicly accessible documents and expert recommendations. Studies, reports and policy documents published between 1st January 2000 and 31st January 2022 will be included. Two independent reviewers will 1) screen and select studies based on *a priori* inclusion/exclusion, 2) conduct quality assessment using the Mixed Methods Appraisal Tool on included studies and 3) extract data. Inductive thematic analysis (objectives 1 and 2), the SEIPS 2.0 model (objective 2) and the Consolidated Framework for Implementation Research (objective 3) will be used to synthesise data.

relevant data from this study will be made available upon study completion.

**Funding:** The authors received no specific funding for this work.

**Competing interests:** The authors have declared that no competing interests exist.

## Ethics and dissemination

This systematic review will use secondary data sources from published documents, and as such research ethical approval was not required. We will disseminate the findings of this study in a peer-reviewed journal and national/international conference(s).

## Trial registration

**PROSPERO registration number**: CRD42022300691. https://www.crd.york.ac.uk/prospero/display_record.php?ID=CRD42022300691.

## Introduction

The way in which hospital beds are used is constantly changing. Demand on hospitals in developed countries [1] has been influenced by a growing, ageing population with more chronic health problems and rapid advances in healthcare technology [2,3]. To meet this rising demand within a constrained funding envelope, inpatient healthcare delivery has also evolved. These changes include a move to day-case [4] and ambulatory care pathways (care delivered without a traditional inpatient bedbase) [5], strategies to avoid admission to hospital [6], rapid discharge interventions including to intermediate care facilities (such as respite, reablement or rehabilitation services outside of an acute hospital) [7] and centralisation of services [8]. The number of inpatient beds per population varies greatly globally, even in developed countries [9], driving innovation for alternative healthcare technologies. Despite the introduction of these changes, the hospital system continues to experience staffing shortages, insufficient funding, inadequate space and deteriorating estate, outdated and insufficient provision of IT, falling bed numbers, long waits and waiting lists.

The COVID-19 pandemic has further highlighted and exacerbated the challenges that global healthcare has been facing. Staff were redeployed and procedures cancelled [10], creating a backlog of care and impacting on healthcare professional training [11]. At the same time, COVID-19 highlighted inequities in care provision [12] and the importance of investing in staff welfare [9,13]. As the global health systems face unprecedented pressures, there is recognition that the healthcare system needs to learn, be able to flex and adapt so that it can respond effectively now and in the future. The COVID-19 context has therefore created opportunities to explore and accelerate improved and innovative care models that release inpatient bed capacity in a safe and effective manner [14]. Examples include enlisting the help of the private sector, rapidly evolving digital health technology [15] and providing hospital level care at home [16].

The recent challenges have accelerated the emergence and development of Hospital at Home (H@H) programmes across the globe [12]. H@H is a treatment model that delivers acute healthcare treatment in the patient's usual place of residence. This model of care has existed in isolated pockets worldwide for several decades, however there is significant variation between how they are rganized and delivered. Studies have also demonstrated similar mortality rates between those admitted to hospital and those treated at home [17]. The additional benefits include a reduction in those needing long term residential care following acute illness and reduction in delirium [17]. The key services that unite these differing models is that they provide some element of acute care that under a traditional healthcare model would have been administered as an inpatient in an acute hospital setting.

There is no internationally recognised definition of H@H, however within the UK, the UK Hospital at Home society released the following consensus statement [18]:

> "Hospital at home provides intensive hospital level care for acute conditions that would normally require an acute hospital bed, in a patient's home for a short episode through multidisciplinary healthcare teams".

Other services from around the world have similar explanations of their services describing a hospital level intervention. In the UK these have often been termed virtual wards, and have received increasing support from the UK government through NHS England to roll out more comprehensive virtual wards delivering H@H interventions [19]. In the USA, an Acute Hospital Care at Home waiver announced during the pandemic [20] has allowed patients to receive at home care and providers gain reimbursement through their insurance programmes in some parts of the country.

Central to the delivery of the hospital level care is the ability to deliver medications within the patient's home. This is critical in order to act in a timely fashion to the patient's condition, clinical assessment and point of care diagnostics. This includes medications traditionally restricted to a hospital setting such as intravenous therapies and oxygen. Currently there is no evidence supporting the delivery of medicines and medicines management within the H@H model. With a wide variation in definition and delivery models of H@H services worldwide, describing these will assist with the understanding of services to support the development of future H@H programmes in the current climate.

## Aim and objectives

Therefore, this systematic review aims to answer how medication management within H@H services established worldwide is conceptulised, described and implemented in practice. The objectives of this review are:

- To describe how medication management in H@H is conceptulised

- To describe and identify key components of medication management in H@H

- To describe and identify variability in the implementation of medication management services within H@H models

## Method

### Protocol and registration

This systemic review is registered in the International Prospective Register of Systematic Reviews (PROSPERO), registration number: CRD42022300691. We report this protocol following the Preferred Reporting Items for Systematic Review and Meta-Analysis Protocols (PRISMA-P) statements [21].

### Eligibility criteria

We will include published studies of any study design, reports and policy documents written on and between 1st January 2000 (this was when the concept of H@H was first introduced but the concept has changed over the years) and 30th November 2022 in the English language that meets one or more of the review objectives. Criteria specifically related to PI(E)COS are as presented in Table 1.

**Table 1. Eligibility criteria for selecting studies.**

|  | Eligibility |
|---|---|
| Participants/population | anyone receiving hospital care meeting the UKH@H society H@H definition at their usual place of residence. |
| Interventions(s), Exposure(s) | medication management in a H@H setting |
| Comparator |  |
| Outcomes | Any |
| Study Design | Any |

**Information sources and search strategy.** The following international electronic databases will be used: Pubmed, Cochrane, grey literature, Web of Science, Cinahl, Medline, organisation websites/resources such as NHS England and Hospital at Home User Group, references of references and experts' recommendations. Table 2 shows the search strategy, including the search terms.

**Study selection.** The search results will be collated on a web based systematic review tool (https://www.rayyan.ai/), and duplicates removed. Independent screening of titles and abstracts will be conducted by two researchers applying pre-specified inclusion and exclusion criteria. Where there are disagreements about eligibility of papers, a third reviewer will assess the paper a consensus method used to determine inclusion. Full-text articles of remaining references will then be obtained and screened independently by two researchers using the same inclusion/exclusion criteria and any disagreements will be resolved by discussion to achieve consensus.

**Data extraction.** The following information will be extracted: authors, year of publication, country where study was conducted (empirical articles) or researchers were based (non-empirical articles), aim of article, descriptions of medication management, key components of medication management model and how medication management has been implemented, where

**Table 2. Search strategy.**

| # | Search terms |
|---|---|
| 1 | "hospital at home" |
| 2 | "hospital@home" |
| 3 | "hospital in the home" |
| 4 | "virtual ward" |
| 5 | "admission avoidance" |
| 6 | 1 or 2 or 3 or 4 or 5 |
| 7 | Pharmacist or clinical pharmacist |
| 8 | pharmac* |
| 9 | medication |
| 10 | "medication management" |
| 11 | "drug delivery" |
| 12 | "medicine optimization" |
| 13 | 7 or 8 or 9 or 10 or 11 or 12 |
| 14 | 6 AND 13 |
| 15 | Limit 14 to English language |

Searches will use Index Terms unique to each database and a combination of Boolean (AND/OR) keywords, as relevant. Grey literature will be identified using Google Scholar.

relevant. For empirical studies, study design and methodology, study setting, sample size, analytical approach used and main findings will also be extracted. A second reviewer will independently extract data from a sample of studies to ensure consistency in the process.

**Risk of bias.** Two independent reviewers will use the Mixed Methods Appraisal Tool (MMAT) version 2018 [22] to critically appraise eligible studies available online at http://mixedmethodsappraisaltoolpublic.pbworks.com/w/file/fetch/127916259/MMAT_2018_criteria-manual_2018-08-01_ENG.pdf. This tool allows the appraisal of quantitative, qualitative and mixed methods studies to reduce the risk of reviewer bias. Any disagreements will be resolved by the inclusion of a third independent reviewer and discussion to achieve consensus.

**Data synthesis.** For objective 1 (to describe how medication management in H@H is conceptulised) and objective 2 (to describe and identify key components of medication management in H@H), all sections of eligible papers will be read and coded inductively using thematic synthesis [23]. In addition, to meet objective 2, the Systems Engineering Initiative for Patient Safety (SEIPS) 2.0 model [24] will be used to code textual data deductively. The SEIPS 2.0 model is a generic system model that shows the elements of a work system and types of work processes that may be required to produce a range of outcomes for different stakeholders. Analyses based on the SEIPS 2.0 model will add to the inductive analysis, to close any potential gaps in our interpretation of findings reported in eligible papers.

Objective 3 focuses on the complex area of implementation of interventions and in this study, the implementation of medication management services within H@H models globally. Therefore, the Consolidated Framework for Implementation Research (CFIR) will be used to code data deductively from eligible studies [25]. The CFIR constructs includes intervention characteristics (SEIPS 2.0 and the thematic synthesis will describe the components of the intervention i.e. medication management, and therefore different from this construct), outer and inner settings, characteristics of individuals using the intervention and process of implementation.

## Ethics and dissemination

This systematic review will use secondary data sources from published documents, and as such research ethical approval was not required. We will disseminate the findings of this study in a peer-reviewed journal and national/international conference(s). The data used in the study are available from the corresponding author on request.

## Patient and public involvement

The systematic review will be based on published data and no patients or members of the public were/will be involved in the design of the study, interpretation or dissemination of the findings.

## Discussion

The H@H concept is a rapidly evolving healthcare service particularly in the UK where the implementation of virtual wards has been a key policy in the NHS post covid-19 recovery plans. It is viewed as an option to deliver healthcare without increasing demand on the traditional inpatient hospital setting. As medication delivery is critical to the H@H function, understanding the literature available will support implementation of future H@H services.

This review will draw on literature from a variety of healthcare systems worldwide. This will not always be applicable to the UK healthcare system and therefore will pose a limitation to the findings. Despite the increase in virtual wards within the UK, not all of these meet the

UKH@H society definition. Only publications of services that meet this definition will be included within this review.

## Supporting information

**S1 Checklist.**
(DOCX)

## Author Contributions

**Conceptualization:** Sophie McGlen, Clare Crowley, Daniel Lasserson, Rosemary H. M. Lim.

**Methodology:** Sophie McGlen, Clare Crowley, Daniel Lasserson, Zahra A. L. Qamariat, Rosemary H. M. Lim.

**Supervision:** Rosemary H. M. Lim.

**Writing – original draft:** Sophie McGlen, Clare Crowley, Zahra A. L. Qamariat, Rosemary H. M. Lim.

**Writing – review & editing:** Sophie McGlen, Clare Crowley, Daniel Lasserson, Zahra A. L. Qamariat, Rosemary H. M. Lim.

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
