## [Decision Letter · Decision Letter 0]

28 Nov 2022

PONE-D-22-13217Hospital at home: a systematic review of how medication management is conceptualised, described and implemented in practice – a study protocolPLOS ONE

Dear Dr. Lim,

Thank you for submitting your manuscript to PLOS ONE. After careful consideration, we feel that it has merit but does not fully meet PLOS ONE’s publication criteria as it currently stands. Therefore, we invite you to submit a revised version of the manuscript that addresses the points raised during the review process.

We look forward to receiving your revised manuscript.

Kind regards,

Ernesto Iadanza

Academic Editor

PLOS ONE

Journal Requirements:

Reviewers' comments:

Reviewer's Responses to Questions

**Comments to the Author**

1. Does the manuscript provide a valid rationale for the proposed study, with clearly identified and justified research questions?

Reviewer #1: Yes

Reviewer #2: Yes

2. Is the protocol technically sound and planned in a manner that will lead to a meaningful outcome and allow testing the stated hypotheses?

Reviewer #1: Yes

Reviewer #2: Partly

3. Is the methodology feasible and described in sufficient detail to allow the work to be replicable?

Reviewer #1: Yes

Reviewer #2: No

4. Have the authors described where all data underlying the findings will be made available when the study is complete?

Reviewer #1: No

Reviewer #2: Yes

5. Is the manuscript presented in an intelligible fashion and written in standard English?

Reviewer #1: Yes

Reviewer #2: Yes

6. Review Comments to the Author

You may also provide optional suggestions and comments to authors that they might find helpful in planning their study.

Reviewer #1: in order to answer to this question: "Have the authors described where all data underlying the findings will be made available when the study is complete?" can authors explain this point?

Reviewer #2: The study protocol is clearly described, as well as the background of the study. However, some sections (eligibility criteria, search strategy, and risk of bias) are poorly explained. They need to be further clarified, maybe by employing tables and/or figures. Full search query should be also provided for reproducibility. Moreover, a final discussion section should be provided, highlighting advantages of conducting a systematic review on H@H topic, major limiting factors, and expectations.

7. PLOS authors have the option to publish the peer review history of their article (what does this mean?). If published, this will include your full peer review and any attached files.

Reviewer #1: No

Reviewer #2: **Yes: **Alessio Luschi

---

## [Author Response · Author response to Decision Letter 0]

15 Dec 2022

Please see our "Response to reviewer comments" document for our comprehensive response.

---

## [Editor Report · Decision Letter 1]

5 Jan 2023

Hospital at home: a systematic review of how medication management is conceptualised, described and implemented in practice – a study protocol

PONE-D-22-13217R1

Dear Dr. Lim,

We’re pleased to inform you that your manuscript has been judged scientifically suitable for publication and will be formally accepted for publication once it meets all outstanding technical requirements.

Kind regards,

Ernesto Iadanza

Academic Editor

PLOS ONE

Additional Editor Comments (optional):

Reviewers' comments:

<quillbot-extension-portal></quillbot-extension-portal>

---

## [Editor Report · Acceptance letter]

18 Jan 2023

PONE-D-22-13217R1 

Hospital at home: a systematic review of how medication management is conceptualised, described and implemented in practice – a study protocol 

Dear Dr. Lim:

I'm pleased to inform you that your manuscript has been deemed suitable for publication in PLOS ONE. Congratulations! Your manuscript is now with our production department. 

Kind regards, 

on behalf of

Dr. Ernesto Iadanza 

Academic Editor

PLOS ONE